# An Approach to Size Sub-Wavelength Surface Crack Measurements Using Rayleigh Waves Based on Laser Ultrasounds

**DOI:** 10.3390/s20185077

**Published:** 2020-09-07

**Authors:** Haiyang Li, Qianghua Pan, Xiaotong Zhang, Zhiwu An

**Affiliations:** 1Key Laboratory of Information Detection and Processing, North University of China, Taiyuan 030051, China; 2China Special Equipment Inspection and Research Institute, Beijing 100029, China; pqh123@163.com; 3Key Laboratory of Advanced Manufacturing Technology, North University of China, Taiyuan 030051, China; zxt_904@163.com; 4Institute of Acoustics of the Chinese Academy of Sciences, Beijing 100190, China; anzhiwu@mail.ioa.ac.cn

**Keywords:** ultrasonic laser, sub-wavelength surface crack, critical frequency

## Abstract

In this paper, the interaction of a broadband Rayleigh wave generated by a laser and an artificial rectangular notch is analyzed theoretically and experimentally. For the theoretical analysis, a Gaussian function is adopted to analyze the modulation of notch depth on the frequency spectrum via reflection and transmission coefficients. By the finite element method, the Rayleigh wave generated by pulsed laser beam irradiation and its scattering waves at cracks are calculated. A curve with a slope close to 4 fitted by crack depth and critical wavelength of the threshold phenomenon is obtained by the wavelet transform and Parseval’s theorem according to simulated and experimental results. Based on this relationship, the critical frequency at which the threshold phenomenon happens due to energy transformation of transmission/reflection Rayleigh waves is adopted to determine the size of sub-wavelength surface crack. The experimental results of artificial notch depth estimation on aluminum alloy specimens consistent with theoretical analysis validates the usefulness of the critical frequency method based on a broadband Rayleigh wave generated by laser ultrasonic.

## 1. Introduction

Without prompt detection, sub-wavelength surface cracks can develop into open and deep cracks, ultimately resulting in structural fracture failure. Quantitative determination of sub-wavelength surface cracks in the aerospace, aviation, and other fields is important to effectively evaluate mechanical properties. Rayleigh waves, whose energy penetration depth is approximately one wavelength during their propagation, are ideally suitable for detecting surface cracks, especially cracks with depths less than the incident Rayleigh wavelength.

The reflected, transmitted, and diffracted Rayleigh waves, whose time and frequency characteristics provide a substantial amount of crack size information, are generated by the interaction between the surface crack and the acoustic wave. During detection, contact transducers in the conventional detection technique are main method to generate and receive Rayleigh waves. Compared to incident wave wavelength, detected cracks are divided into two kinds that are long cracks with depths greater than at least one wavelength and sub-wavelength cracks whose depth is less than the incident wavelength. For the detection of long surface cracks, the pitch-catch [1] and pulse-echo method [2] which adopt the propagation characteristics of acoustic waves, reflection and transmission coefficients [3] employing amplitude attenuation are usually applied. These linear acoustic theoretical methods work due to the fact the depth of the surface crack is deep enough leading to the propagating Rayleigh wave being seen as a ray. However, for sub-wavelength surface crack detection, a more complex measurement mechanism considering boundary condition of crack edge is adopted. Because the sub-wavelength crack dimensions are smaller than the elastic wavelength longitudinal component of the acoustic field generated by Rayleigh waves around crack surfaces parallel to the direction of propagation and approximately uniform and the other stress components are approximately equal to zero, Auld [4] developed an analytical method based on reflection coefficients to estimate sub-wavelength surface crack sizes. Resch [5] determined the actual acoustic parameters of transducers using an analytical equation in [4] and experimentally applied this measurement method to infer crack depth, crack growth, and crack opening behavior during fatigue cycling. Subsequently, Hassan [6] used finite element simulations to analyze the relationship between surface crack depth and reflection coefficients presented in [4]. Based on a hypothesis in which surface cracks are considered energy absorbers when the crack depth satisfies a multiple of a half-wavelength long resulting in extreme points on the reflection coefficient vs. frequency curve, Domarkas [7] proposed an analytical expression to estimate crack depth. In addition, a nonlinear interaction between Rayleigh wave and surface crack is simulated by Yuan [8] using the finite element method to evaluate sub-wavelength crack depths. How a wave-amplitude and frequency-dependent response can be physically relevant for a certain frequency range is studied by [9,10]. Thus size detection of sub-wavelength cracks is a more complex problem due to the fact its size is smaller than the incident Rayleigh wave’s wavelength. For this reason, it is necessary to develop a new method combining both reflection and transmission Rayleigh waves for size detection of sub-wavelength cracks.

To overcome the limitations of the detection environment and specimen size due to the fact traditional measurement methods are dependent on contact transducers for acoustic signal generation and detection, Rayleigh waves based on ultrasonic laser with high resolution, wide broadband, and non-contact detection have been developed to investigate surface cracks [11,12,13,14,15,16]. In these publications, the characteristics of reflected and transmitted Rayleigh waves at cracks are analyzed to accurately evaluate crack depth. Cooper [11] and Hess [12] investigated reflection and transmission Rayleigh waves at surface cracks. Using the frequency spectrum of reflection and transmission acoustic waves, Cooper studied the effect of surface cracks on the frequency spectrum and arrival time of reflection and transmission Rayleigh waves. Finally, he created a crack depth evaluation methodology using the arrival time of reflected Rayleigh waves and mode-converted transverse waves at the base of cracks. Hess adopted an experimental design of generation and reception positions at a fixed distance to detect surface cracks. The reflection coefficient function based on reflection waves and the time of flight based on transmission waves were compared to acquire acoustic wave signals in different zones on two sides of cracks. At the singular tip of cracks, scattering waves and mode conversion at the cracks were observed by simulation and experimental methods in the time domain to estimate surface crack depth [11,13,14]. A conformal mapping method that solves the limitation of incident wavelength and distance is proposed to evaluate crack depth using the transmitted Rayleigh waves [15]. Taking into account the contact interface stiffness of closed cracks, Lomonosov [16] applied the reflection coefficient function with respect to frequency of Rayleigh wave based on ultrasonic laser to estimate the size of closed surface cracks. For an artificial surface crack, Zhou [17] pointed out that the transmission coefficients of Rayleigh waves generated by laser at crack are dependent on their frequency range. The sensitivity and reliability of crack depth characterization is controlled by the Rayleigh wave wavelength and bandwidth using an ultrasonic laser to generate and detect Rayleigh waves. Tournat [18] investigated a nonlinear mixed frequency method to quantitatively evaluate crack depth by using two waves generated by a laser having low and high frequency leading to a more sensitive image for crack achieved. In addition, a near field method called Scanning Laser Source has been extensively studied by lots of researchers [19,20,21]. In this method, the arrival time of a Rayleigh wave and its scattering waves are analyzed to calculate for crack depth evaluation when the incident laser just impacts on the crack. Overall, most obtained achievements focus on long crack detection and few studies on the sub-wavelength surface crack detection are developed using Rayleigh waves based on lasers.

A quantitative determination method for sizing sub-wavelength surface cracks by the threshold phenomenaon’s critical frequency is proposed in this study. A broadband Rayleigh wave is the combined result of individual components with different frequencies, whose energy is concentrated on the surface medium in one wavelength of each component. When they encounter a crack during their propagation, low-frequency acoustic waves more easily travel through the crack, while high-frequency acoustic waves are more likely be reflected [12]. Due to the different reflection and transmission capacities of individual components in a range of broadband frequencies, a threshold phenomenon of energy transformation of reflection and transmission Rayleigh waves in the frequency domain is analyzed. Above the threshold phenomenon’s critical frequency, the energy of reflection Rayleigh wave is greater than that of the transmission Rayleigh wave during acoustic scattering at the crack. However, below the threshold phenomenon’s critical frequency, the energy of reflection Rayleigh wave is less than that of the transmission Rayleigh wave. The threshold phenomenon’s critical frequency is related to crack depth. In the theoretical part, a Gaussian function is employed to describe the modulation of crack size on the reflection and transmission coefficients. Both amplitude reduction and center frequency shift are observed in the frequency spectrum. Using the finite element method (FEM), the acoustic field including the reflection and transmission Rayleigh wave at surface crack is obtained. Using the wavelet transform and Parseval’s theorem, critical frequencies at cracks with different depth are recorded by analyzing reflection and transmission Rayleigh waves in the frequency domain. A fitted curve with a slope of close to 4 is obtained by critical wavelength vs. crack depth. The crack depth is successfully estimated using threshold phenomenon’s critical frequencies. The experimental results of tests conducted on artificial notches agree with the simulation results. Finally, a discussion and summary conclude this paper.

## 2. Theoretical Analysis and Signal Processing Method

### 2.1. The Effect of Reflection and Transmission Coefficients on the Frequency Spectrum

In this study, the reflected and transmitted Rayleigh waves generated by the interaction of an incident broadband Rayleigh wave impacted by a pulse laser and surface crack are used to estimate crack depth size. The acoustic field containing reflected, transmitted and scattering Rayleigh waves at the surface crack is shown in Figure 1.

The incident Rayleigh wave’s displacement is represented by the plane *x*_1_–*x*_2_, where axis *x*_1_ is the propagation direction and axis *x*_2_ is vertical to the propagation direction and points into the medium as follows:*u*(*x*_1_, *x*_2_) = *AD*(*x*_2_)exp(*ik**x*_1_–*wt*), *i* = 1, 2(1)
where *D*(*x*_2_) is an exponential decay function with respect to the vertical distance away from the sample’s surface. A is the amplitude of the incident Rayleigh wave propagating along axis *x*_1_. The reflected Rayleigh wave returns along axis -*x*_1_ in the opposite direction to that of the incident wave, while the transmitted Rayleigh wave goes through the crack along axis *x*_2_ in the same direction as the incident wave. *R*^r^ and *R*^t^ are defined as reflection and transmission coefficients, respectively, so the reflected and transmitted Rayleigh waves are represented as:*u*^r^_i_(*x*_1_, *x*_2_) = *R*^r^*A*_i_*D*(*X*_2_)exp(–*ik**x*_1_–*wt*), *i* = 1, 2(2)
*u*^t^_i_(*x*_1_, *x_2_*) = *R*^t^*A*_i_*D*(*X*_2_)exp(*ik**x*_1_–*wt*), *i* = 1, 2(3)

To analyze frequency characteristics, a Gaussian function with a center frequency of *w*_0_ = 2 MHz is used to represent a broadband acoustic wave in the frequency spectrum as follows:*s*(*w*) = exp( − (*w* − *w*_0_)^2^/(*w*_0_/2)^2^)(4)

Each component is obtained using a Dirac function to extract the individual frequency component of an acoustic signal in a broadband range as follows:(5)$$\sum_{m=1}^{n}s\left(\omega_{m}\right)=\sum_{m=1}^{n}s\left(\omega \right)\delta \left(\omega_{m}\right)$$
where *w*_m_ is the frequency of individual component and *s*(*w*) is the acoustic signal. Substituting Equation (1) into Equation (5) as an acoustic signal gives the incident wave:(6)$$u^{i}=A_{d}H$$
where *u* = [*u*_1_(*x*_1_, *x*_2_, *w*_1_), *u*_2_(*x*_1_, *x*_2_, *w*_2_)...*u*_m_(*x*_1_, *x*_2_, *w*_m_)]^T^ is the displacement of incident acoustic wave and the angle frequencies satisfy *w*_1_ < *w*_2_ <....*w*_m_. *A*_d_ is a triangular matrix whose diagonal elements are composed of the product of the individual component’s amplitude and depth function defined as *A*_d_ = [*A*_1_*D*_1_(*x*_2_), *A*_2_*D*_2_(*x*_2_)...*A*_m_*D*_m_(*x*_2_)]^T^. *H* is a vector composed of the propagation functions *H* = [exp(*ik**x*_1_-*w*_1_*t*), exp(*ik**x*_1_-*w*_2_*t*)...exp(*ik**x*_1_-*w*_m_*t*)]. The reflected and transmitted waves are also represented using this method as follows:(7)$$u^{r}=R^{r}A_{d}H$$
(8)$$u^{t}=R^{t}A_{d}H$$
where *R*^r^ and *R*^t^ are two vectors composed of reflection and transmission coefficients at different frequencies defined by *R*^r^ = [*R*^r^(*w*_1_), R^r^(*w*_2_)...R^r^(*w*_m_)] and *R*^t^ = [*R*^t^(*w*_1_), *R*^t^(*w*_2_)....*R*^t^(*w*_m_)], which denote the reflection and transmission coefficients of each frequency component. In Equation (5), the reflection and transmission coefficients have an increasing or decreasing trend with the frequency. Two functions of *R*^r^ = *w*^1/2^ and *R*^r^ = *w* for reflection coefficients and two functions of *R*^t^ = −*w*^1/2^ + 1 and *R*^t^ = −*w* + 1 for the transmission coefficients are adopted satisfying *R*^r^ and *R*^t^ less than one and positive, and the sum of *R*^r^ and *R^t^* equals one. *R*^r^ = *w* and *R*^t^ = −*w* + 1 are the reference functions. 

The reflection and transmission coefficients vs. frequencies are shown in Figure 2a,c, and the effect of the reflection and transmission coefficients on the spectrum of reflection and transmission Rayleigh waves are shown in Figure 2b,d. According to Figure 2b, the center frequencies, which are the corresponding frequencies with the maximum amplitudes, increase because low-frequency reflection coefficients are lower than at high frequencies in the reflection Rayleigh wave shown in Figure 2a. However, as demonstrated in Figure 2d, the center frequencies decrease because low-frequency transmission coefficients are higher than at high frequencies in the transmission Rayleigh wave shown in Figure 2c. The spectrum’s center frequency shift actually modulates crack depth applied to the frequency spectrum by reflection and transmission coefficients. The reflection coefficients of the function *R*^r^ = *w*^1/2^ are much closer to experimental data than the results in [13] with the higher values than the function *R*^r^ = *w* resulting in the higher amplitude of reflection Rayleigh wave in the frequency spectrum. Because the sum of the reflection and transmission coefficients is one, the corresponding transmission coefficient function for reflection coefficients function *R*^r^ = *w*^1/2^ is *R*^t^ = −*w*^1/2^ + 1, which has lower values than function *R*^t^ = −*w* + 1, resulting in a higher amplitude reduction in the frequency spectrum. The crack depth’s effect on the frequency spectrum of reflection and transmission Rayleigh waves was studied in [11], where the reflected and transmitted Rayleigh amplitudes showed a rapid reduction at special frequencies, whose corresponding wavelength depend on crack depth.

### 2.2. The Transmission/Reflection Threshold Phenomenon

For acoustic waves with a single frequency incident to the crack whose energy is *E_i_*, the reflected, transmitted, and scattered Rayleigh waves are generated at the crack whose energies are *E*_r_, *E*_t_, and *E*_s_, respectively, and satisfy *E*_i_ = *E*_r_ + *E*_t_ + *E*_s_. In this work, depth of detected crack is smaller than that of incident wavelength, so scattering waves at the crack tip are ignored. Therefore, the sum of reflection and transmission waves energy at the crack is constantly equal to incident wave energy. Among reflection waves, high-frequency acoustic waves are more likely to reflect than low-frequency acoustic waves. But the situation is opposite for transmission waves. Thus, a threshold exists due to energy transformation of reflection and transmission waves at cracks as follows:(9)$$\left\{ \begin{array}{l} E_{r}^{}\left(\omega \right)<E_{t}\left(\omega \right),l>l_{0}\\ E_{r}\left(\omega \right)>E_{t}\left(\omega \right),l<l_{0} \end{array}\right.$$
where *l* is the ratio of the wavelength λ of the incident wave to the depth *h* of the surface crack, that is, *l* = λ/*h*, and *l*_0_ is the critical value of the energy transformation of transmission and reflection waves.

According to Equation (9), when the ratio *l* defined by wavelength λ and crack depth *h* is higher than *l*_0_, the energy of reflection wave *E*_r_ is less than the energy of transmission wave *E*_t_. When the ratio *l* is smaller than *l*_0_, the energy of reflection wave *E*_r_ is higher than the energy of reflection wave *E_t_*. That is, there is a critical value *l*_0_ at which the transmitted and reflected wave energy transforms. The threshold phenomenon’s corresponding wavelength is called critical wavelength, and the corresponding frequency is called critical frequency. The relationship between critical wavelength and critical frequency is calculated using λ = *c*/*f.* The key to measuring the crack depth is determining the critical frequency of threshold phenomenon of the transmission/reflection energy transformation.

### 2.3. Wavelet Transform and Parseval’s Theorem

In Figure 3, an original signal with three frequency components at 5, 18 and 38 Hz is decomposed into three signals called signal 1, signal 2, and signal 3. The frequency range of signal 1 is 7.5 MHz–12.5 MHz, the frequency range of signal 2 is 12.5 MHz–25 MHz, and the frequency range of signal 3 is 25 MHz–50 MHz. The wavelet transform decomposes a broadband Rayleigh wave’s acoustic signals into individual frequency components as demonstrated in Equation (5).

To obtain the energy of decomposed signals, Parseval’s theorem in the frequency spectrum is adopted as follows:(10)$$E_{k}\left(\omega \right)={\left|\sum_{m=1}^{n}s_{k}\left(\omega_{m}\right)\right|}^{2}$$
where *E*_k_(*w*) is the energy of the acoustic signal and k is the decomposed signals. The individual frequency *s*(*w*_m_) is a narrow band signal obtained via the wavelet transform method, not by a Dirac function as in Equation (4). Using Equation (10), the energy of reflected and transmitted acoustic signals shown in Figure 2 are obtained.

According to the results in Figure 4, the surface crack effects not only the frequency spectrum but also the energy distribution of acoustic signals. In Figure 4, the reflection acoustic wave’s energy has a higher value and the transmission acoustic wave’s energy has lower value for the function *R*^r^ = *w*^1/2^ and *R*^t^ = −*w*^1/2^ + 1, respectively. The energy of both reflection and transmission signals depends on the frequency spectrum, which is modulated with crack depth by reflection and transmission coefficient functions. To ascertain the critical frequency of reflection/transmission conversion in Equation (9) using the wavelet transform, the reflected and transmitted Rayleigh waves are decomposed into signals in the different frequency range and then combined with Parseval’s theorem in Equation (10) to analyze the relationship between crack depth and critical wavelength.

## 3. FEM Simulation

### 3.1. Simulation Settings and Center Frequency Shift

In FEM simulations using COMSOL software, heat transfer solids, solid mechanics, and their multi-physics including thermal expansion and temperature coupling are employed to calculate stress perturbations generated by laser irradiation. When a micro surface crack in a sample is parallel to laser source and vertical to the sample surface, this is a two-dimensional problem. The line source model in a thermoelastic system is described as a product of two Gaussian functions, which are temporal function with a spatial width of 100 um and spatial function with a time duration of 10 ns, respectively. Aluminum with Rayleigh wave velocity of 2948 m/s is used as the simulation medium because the experimental sample is made from the same material. 

A geometric model with a length of 30 mm and width of 5 mm whose right, left, and lower boundaries are set as low refection boundary conditions is meshed with free triangular elements of the minimum size of 6e–4 mm and the maximum size of 0.3 mm. A simulation time of 8 us with 30 ns steps is adopted throughout the simulation to obtain the ultrasonic field generated by the pulse laser beam as shown in Figure 5a. The reflected and transmitted Rayleigh waves with considerable size information on rectangular notches are shown in Figure 5b. The convergence curve for this simulation is presented in Figure 5c.

Although the ultrasonic field contains many types of waves, Rayleigh wave is suitable for surface crack detection due to its propagation along the sample’s surface compared with other wave types generated. Rectangular notches with widths of 0.2 mm, 0.3 mm, 0.4 mm, and 0.5 mm are set on the sample’s surface. The reflected and transmitted waves on two sides of the cracks are recorded in Figure 6a,b, and their frequency spectra are shown in Figure 6c,d.

Each frequency component in a broad range of 0–6 MHz generated by the pulse laser, has a different capacity to reflect and transmit the surface crack, so the reflection and transmission coefficients have surface crack size information. The modulation of crack size on the reflection and transmission Rayleigh waves causes center frequencies to shift in the frequency spectrum leading to changes in the energy distribution in the broadband frequency range. The center frequencies extracted from Figure 2b,d and Figure 6c,d vs. cracks depth are shown in Figure 7.

The comparison between Figure 7a,b shows that center frequencies obtained by using power functions display an increasing and decreasing trend, respectively, similar to the results obtained by the FEM. Combined with an oblique line obtained by center frequencies in Figure 6d, it is seen that center frequency shift of simulated transmitted Rayleigh wave in Figure 7b has a slighter shift than that of power function in Figure 7a. Two reasons for this phenomenon are that power function describing transmission ability exaggerates the effect of crack depth on the transmission coefficients, and in the simulation procedure, sub-wavelength crack is so small that it is not sensitive to the centre frequency component of simulation Rayleigh wave. So we can see that for sub-wavelength crack detection, the measured accuracy of crack depth is limited by transmission wave due to little change of transmission coefficients.

### 3.2. The Analysis of Critical Frequencies

To obtain critical frequencies of reflection/transmission threshold of simulated reflected and transmitted Rayleigh waves at cracks with different depth, the wavelet transform having five decomposed layers is applied. The frequency range of decomposed layers are shown in Table 1. The energy distribution vs. frequency spectrum of different layers at cracks with different depth are presented in Figure 8.

In Figure 8a, from layer d1 to layer d2, reflection waves’ energies are greater than those of transmission signals. However, from layer d3 to layer d5, reflection waves’ energies are lower than those of transmission waves. Therefore, transmission/reflection wave’s threshold phenomenon occurs at a critical frequency of 3.34 MHz extracted from layer d2.

The critical frequencies and wavelengths of threshold phenomenon at crack depth ranging from 0.2 mm to 0.5 mm are shown in Table 2. A fitted curve obtained using critical wavelength and crack depth is shown in Figure 9.

The fitted curve obtained using critical wavelength and crack depth is a line with a slope of 4.02. This means that the critical wavelength that is extracted in the threshold phenomenon of the energy transformation of reflection and transmission Rayleigh waves is related to crack depth. This relationship between the wavelength and crack depth was also studied by Domarkas [7], who reported that the surface crack was a resonator for absorbing the energy of acoustic waves when the crack depth and incident wavelength satisfied the relationship *h* = *λ*/(4 (2 * *n* + 1), where *h* is the crack depth and *λ* is the wavelength. In this study, the threshold phenomenon of energy transformation of reflection and transmission Rayleigh waves occurs when the ratio of critical wavelength to crack depth close to 4, which precisely satisfies the relationship between the crack depth and wavelength proposed by Domarkas when the value *n* equals zero. So the critical frequency method, which is represented by the fitted curve of critical wavelength vs. crack depth, has potential to estimate crack depth.

### 3.3. The Effect of Element Size on the Estimated Results

The element size is an important parameter for acoustic field simulation by the FEM. The maximum size of 0.2 mm and 0.5 mm of free triangle element are adopted to analyze the effect of simulation parameter on the estimated crack depth. The critical frequencies, critical wavelength, estimated cracks depth and error for different simulation parameters are shown in Table 3 and Table 4. The corresponding fitted curves are presented in Figure 10.

According to the fitted curve in the Figure 9 obtained by the maximum size of element of 0.3 mm and two fitted curves in the Figure 10a and b obtained by the maximum size of element of 0.2 mm and 0.5 mm, respectively, we can see that the relationship between critical wavelength and crack depth obtained by different simulation parameters nearly satisfies the same linear slope. So it demonstrates that the simulation results of critical frequency method have good repeatability in this paper under the different element size.

## 4. Experimental Procedures

### 4.1. Experimental Measurement Design

The experimental measurement design consists of an excited part, detection part, and automatic controlling part. The excited part of a CFR200 laser with a pulse width of 20 ns and a repetition frequency of 20 Hz launches a laser beam to form a thermoelastic line source on the sample’s surface using a cylindrical lens with a focal length of 200 mm. The reflected and transmitted Rayleigh waves at the cracks are acquired by a QUARTET 500 mV receiver based on the principle of Michelson’s interferometer. During the experimental procedure, the line source impact position and detection spot are static, while the detected sample is moved using an auto controlling part of LU scanning software with two axes in the horizontal and transverse directions with a total distance of 250 mm × 250 mm and a minimum scanning step of 0.006 mm. A schematic diagram of experimental design is shown in Figure 11a.

A Rayleigh wave is generated and detected on the surface of a sample with a crack. The distance between source line and probe spot is maintained constant at 6 mm as sample moves the distance of 15 mm in steps of 0.05 mm. The scanning path is divided into two parts called zone 1 and zone 2 as shown in Figure 11b. Four aluminum samples with artificial rectangular notches with depth of 0.2, 0.3, 0.4 and 0.5 mm, respectively, at wavelength and notch depth ratios less than one due to the bandwidth of Rayleigh wave generated by the proposed ultrasonic laser design are in a range of 0–5.21 MHZ. The samples’ geometric dimensions are shown in Figure 11c.

### 4.2. B-Scan Images and Width Measurement of Surface Cracks

B-scan line scanning images across the rectangular notches using this experimental design are shown in Figure 12. The reflected, transmitted, and direct signals are labeled in Figure 12a. Zones 1 and 2 in Figure 11b are also labeled in Figure 12a. Acoustic signals in the time domain at a scanning distance of 0.5 mm in zone 2 and a scanning distance of 12.45 mm in zone 1 are shown in Figure 12e,f, respectively.

Using Figure 12a as an example, the direct and transmitted Rayleigh waves are displayed as two parts of one straight line due to the constant distance between the exciting and receiving lasers. The inclined line represents Rayleigh wave reflected from the surface cracks and its slope is equal to Rayleigh wave’s velocity. A cross point forms by the end of the direct and reflection acoustic signals and the start of the transmission acoustic signal, and at this point, the location of surface cracks is easily realized. This is because the cross point corresponds to the position where the receiver laser is just at the notch as specimen moves as shown in Figure 11b.

The transmitted wave at 0.5 mm in zone 2, the direct wave signal, and the reflected wave signal at 12.45 mm in zone 1 are extracted from Figure 12a and shown in Figure 12e,f. Because of mode conversion from the Rayleigh wave at the tip notch, longitudinal and transverse waves are also observed in the B-scan images.

## 5. Results and Analysis

### 5.1. Spectrum Analysis of the Reflection and Transmission Waves at the Surface Cracks

Using the wavelet transform and Parseval’s theorem, the energy curves vs. frequency of reflection and transmission waves in each layer are obtained and shown in Figure 13. The solid red lines represent the energy curves of reflection surface waves, and the solid black lines represent the energy curves of transmission surface waves. These experimental results are also reported in [22]. In addition, the *x* axis represents the frequencies (unit: MHz), and the *y* axis denotes the energy (unit: mJ).

In Figure 13, S is the original signal, and the frequency range of d1–d5 signals are the same as the simulation signals in Figure 9. To ascertain the characteristics of decomposed reflection and transmission signals, decomposed signals (a) in Figure 13 are represented in the frequency-spatial domain in Figure 14.

In Figure 14a,e in the frequency-spatial domain denote d1–d5 decomposed signals, respectively. Because the frequency range of d2–d5 signals are in the bandwidth of Rayleigh wave’s frequency spectrum based on ultrasonic laser used in this study, signals in (a–d) have stronger amplitudes than the other signals in (e). As shown in Figure 13a, in the range of 0.52 MHz–1.04 MHz, 1.04 MHz–2.08 MHz, and 2.08 MHz–4.17 MHz, the Rayleigh waves’ transmission energies are higher than their reflection energies. In the range of 4.17 MHz–8.34 MHz and 8.34 MHz–16.67 MHz, Rayleigh waves’ reflection energies are higher than their transmission energies. Because the bandwidths of Rayleigh waves generated by ultrasonic laser are 0 MHz–5.12 MHz, critical frequencies must be searched in the range of 4.17 MHz–8.34 MHz of the d2 signal, where the threshold phenomenon of energy transformation of the reflection and transmission Rayleigh waves occurs.

By the critical frequency method, the first step is how to decide the number of decomposed layers of the wavelet transform. It is necessary to combine decomposed signals in the frequency-spatial domain and curves of energy vs. frequency to obtain a suitable number of decomposed layers. The bandwidth of incident Rayleigh wave is an important factor to decide decomposed layer number. Too large decomposition frequency width result in missing the critical frequency. In this paper, the bandwidth of incident Rayleigh wave is 0–5.12 MHz and the frequency range of decomposed layers are presented in Table 1. It is seen that d2–d5 signals are in the bandwidth of the incident Raleigh wave. That is to say that the bandwidth of incident Rayleigh wave is divided into four layers. Energy distribution of reflection and transmission Rayleigh wave are obtained by the wavelet transform and Parseval’s theorem. The second step is to determinate the critical frequency according to the theoretical analysis in Section 2.2. The decomposed layer, where the energy transform of reflection and transmission Rayleigh waves happens, is a frequency range. At this layer, the minimum frequency, at which the energy transform of reflection and transmission Rayleigh wave starts to happen, is defined as the critical frequency. Ultimately, crack depth is estimated by a quarter of the critical wavelength. The corresponding critical frequency of critical wavelength have to be in the bandwidth of incident Rayleigh wave. Based on this reason, the maximum depth of surface crack is limited by the bandwidth of Rayleigh wave generated by ultrasonic laser because no signals exist but noise when their frequencies exceed the bandwidth range.

### 5.2. The Wavelet Packet Method to Find the Critical Frequency

To accurately ascertain critical frequencies, the wavelet packet method is adopted to obtain decomposed signals in a narrower decomposed signal band shown in Figure 15. The d2 signal is divided into eight signals (S1–S8) with a uniform frequency range, and the d3 signal is divided into four signals (S9–S12) with a uniform frequency range. The detailed frequency distributions of decomposed signals are shown in Figure 15.

The more detailed decomposed frequencies of d2 and d3 signals are shown in Figure 15a. In the range of 4.17 MHz–8.34 MHz, we find the minimal frequency, at which energy transformation of reflection and transmission Rayleigh waves begins to happen, as the critical frequency. In Figure 15a, the energy of transmission Rayleigh waves is higher than that of reflection Rayleigh waves in S10–S12, and the energy of transmission Rayleigh waves is lower than that of reflection Rayleigh waves in S9. A more accurate critical frequency is obtained in S9 in Figure 15a than in the d2 signal in Figure 13a. According to the results in Figure 13 and Figure 15, the critical frequencies are found by the threshold phenomenon of reflection and transmission energy conversion.

### 5.3. The Critical Frequency Method

The estimated notches depth are calculated using one-quarter of the critical wavelength, which is obtained by equation λ= *c*/*f*, where *f* is the critical frequency. To get more accurate estimated estimated values, three pairs of reflection and transmission Rayleigh waves at different positions are adopted to remove the random noise during detection. In addition, a moving average filter based on weighted linear least squares is used to eliminate noise generated by detection system. Their positions are shown in Table 5. The critical frequencies, critical wavelength, estimated crack depth and error are shown in Table 6, Table 7 and Table 8. The mean values and variance of estimated crack depth are shown in Table 9. A fitted curve of critical wavelength vs. crack depth with statistical information is presented in Figure 16.

According to results in Figure 16, the slope of fitted curve based on experimental data is 3.96, which is close to the simulation results in Figure 9. This demonstrates experimentally the rationality of using the value of 4 to calculate crack depth via the critical frequency method. Broadband Rayleigh waves generated by ultrasonic laser can be used to quantitatively determine crack depths via the critical frequency method using the wavelet transformation and Parseval’s theorem. Compared with other quantitative determination methods for sizing surface cracks using Rayleigh waves based on ultrasonic lasers, which only adopt one kind of acoustic wave of reflection or transmission Rayleigh waves, the critical frequency method employs both reflection and transmission Rayleigh waves to estimate surface crack depth. Thus, the measurement results using the critical frequency may provide more comprehensive information on surface crack size. The results of this study provide a deeper understanding of the interactions of Rayleigh waves generated by ultrasonic lasers and surface cracks.

## 6. Conclusions

Rayleigh waves based on lasers are broad band acoustic waves which contain a great deal of individual components with different frequencies. When they meet a surface crack the individual components display different ability to reflect and transmit at the crack. The modulation of surface crack size on the frequency spectrum of acoustic wave is theoretically analyzed by Gauss and power functions.

A threshold phenomenon related with the ratio of incident wavelength and crack depth exists in the frequency spectrum due to the comprehensive interaction of all frequency components and the surface crack. The critical wavelength of threshold phenomenon related to crack depth has potential to evaluate crack depth.

The wavelet transformation and Parseval’s theorem are adopted to analyze reflection and transmission Rayleigh waves and obtain the critical frequency. Finally, both the simulation and experimental results show that the ratio of critical wavelength of threshold phenomenon to crack depth is close to the value of 4. Owing to this relationship, the critical frequency method using Rayleigh waves based on laser is proposed to size depth of sub-wavelength surface crack.

## Figures and Tables

**Figure 1 sensors-20-05077-f001:**
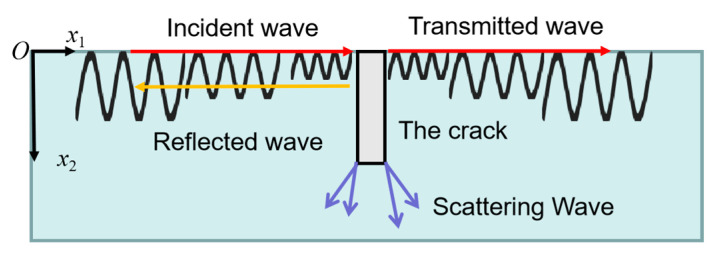
The interaction between an incident Rayleigh wave and the cracks.

**Figure 2 sensors-20-05077-f002:**
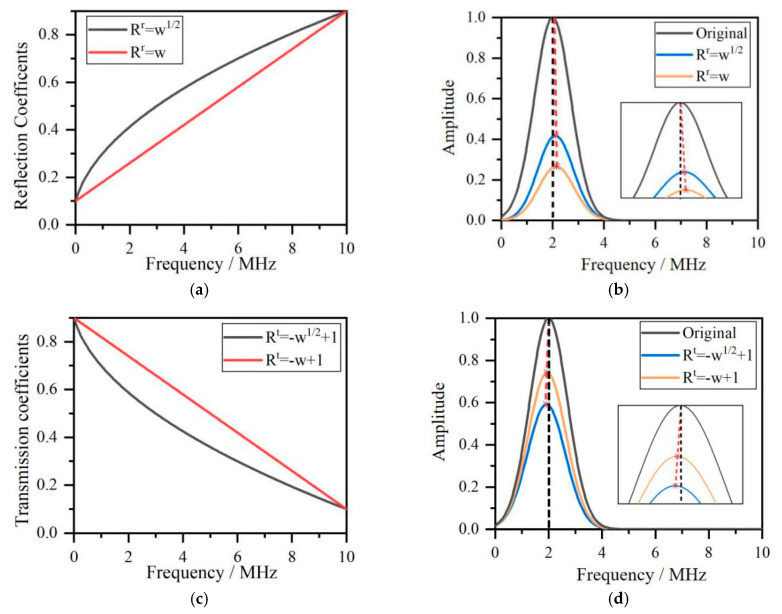
(**a**) Reflection coefficients functions. (**b**) Frequency spectrum of reflection Rayleigh wave. (**c**) Transmission coefficients functions. (**d**) Frequency spectrum of transmission Rayleigh wave.

**Figure 3 sensors-20-05077-f003:**
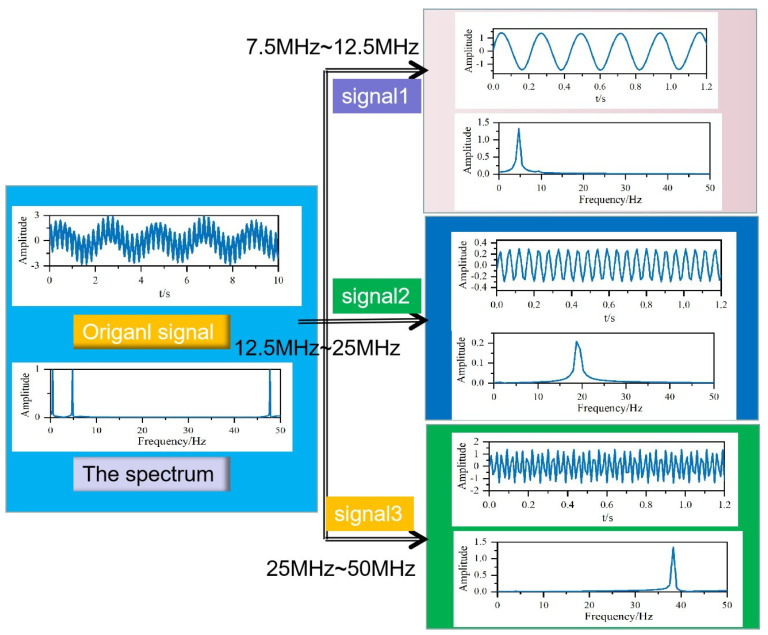
The decomposed signals using the wavelet transform.

**Figure 4 sensors-20-05077-f004:**
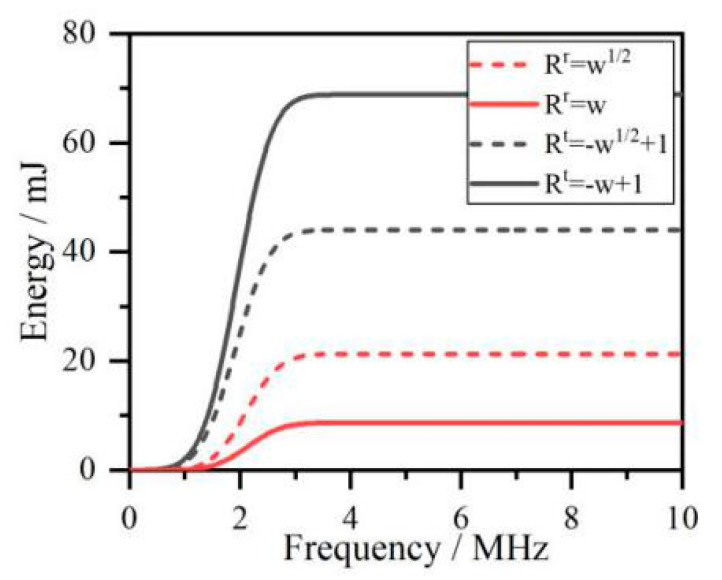
The energy distribution of the acoustic signals.

**Figure 5 sensors-20-05077-f005:**
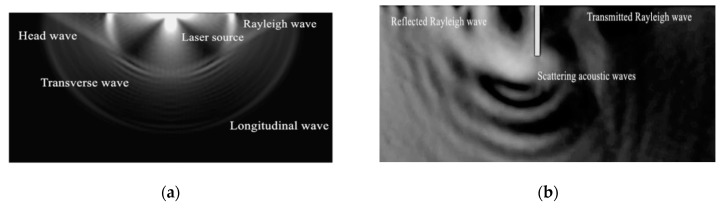
(**a**) Ultrasonic field generated by a laser. (**b**) Scattering waves at the cracks. (**c**) The convergence curve.

**Figure 6 sensors-20-05077-f006:**
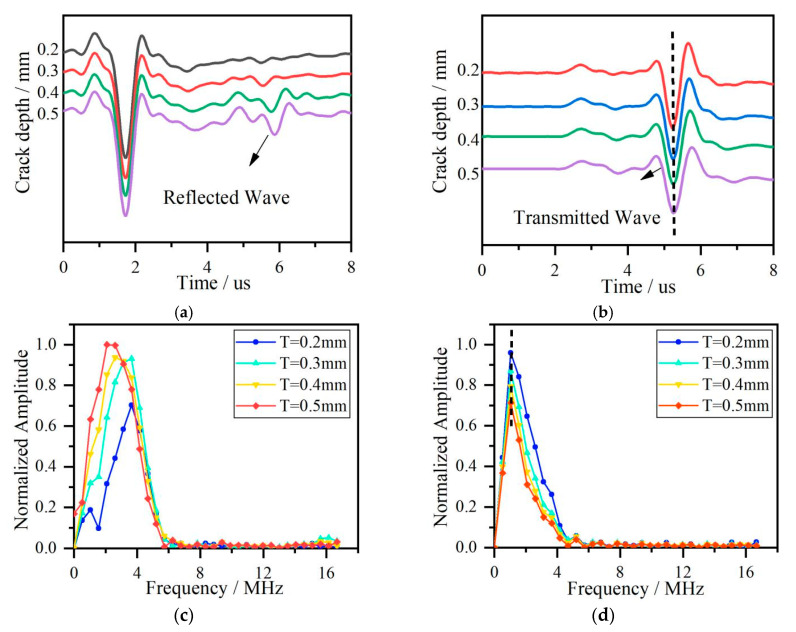
(**a**) Reflected Rayleigh waves. (**b**) Transmitted Rayleigh waves. (**c**) The frequency spectra of the reflected Rayleigh waves. (**d**) The frequency spectra of the transmitted Rayleigh waves. The acoustic waves are obtained at cracks with depth as 0.2 mm, 0.3 mm, 0.4 mm, and 0.5 mm.

**Figure 7 sensors-20-05077-f007:**
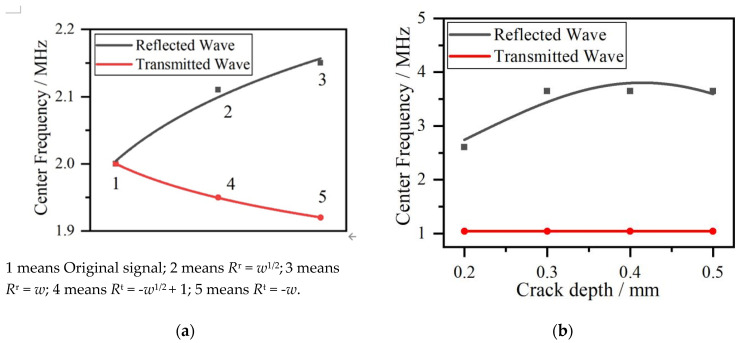
(**a**) The results based on Gaussian function. (**b**) The results based on the FEM.

**Figure 8 sensors-20-05077-f008:**
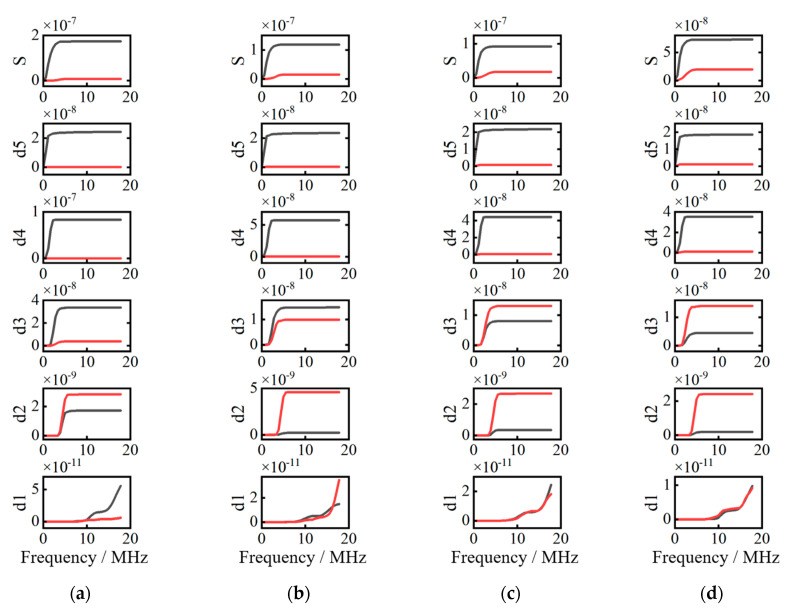
The energy distribution vs. frequency spectrum. (**a**) Crack depth is 0.2 mm, (**b**) crack depth is 0.3 mm, (**c**) crack depth is 0.4 mm, and (**d**) crack depth is 0.5 mm.

**Figure 9 sensors-20-05077-f009:**
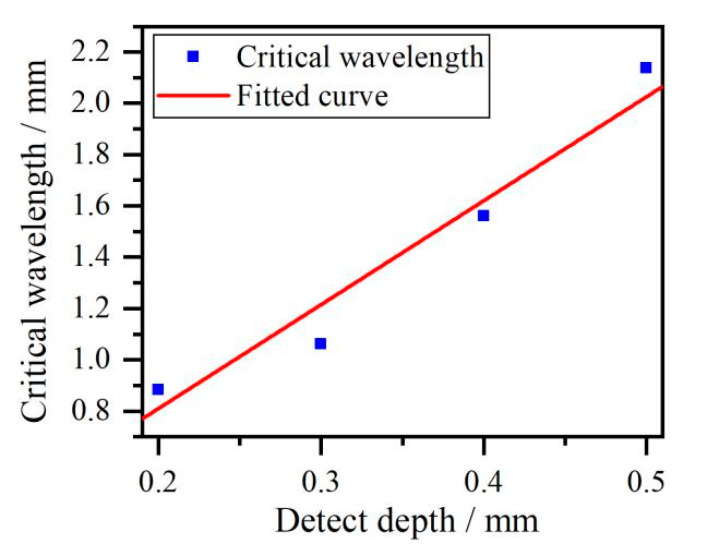
The critical wavelengths and cracks depth.

**Figure 10 sensors-20-05077-f010:**
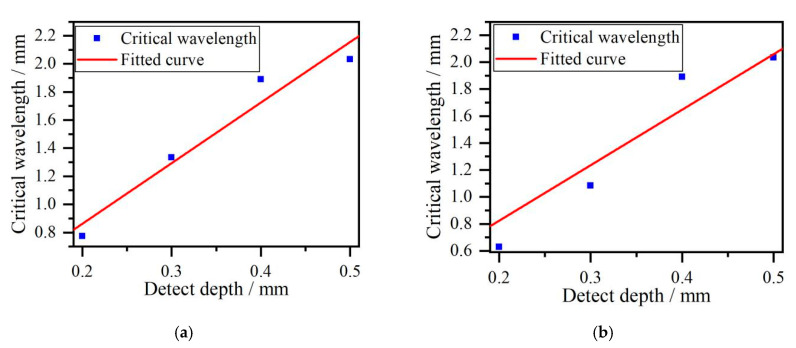
The fitted curves by simulation elements of (**a**) the maximum size 0.2 mm, (**b**) the maximum size 0.5 mm.

**Figure 11 sensors-20-05077-f011:**
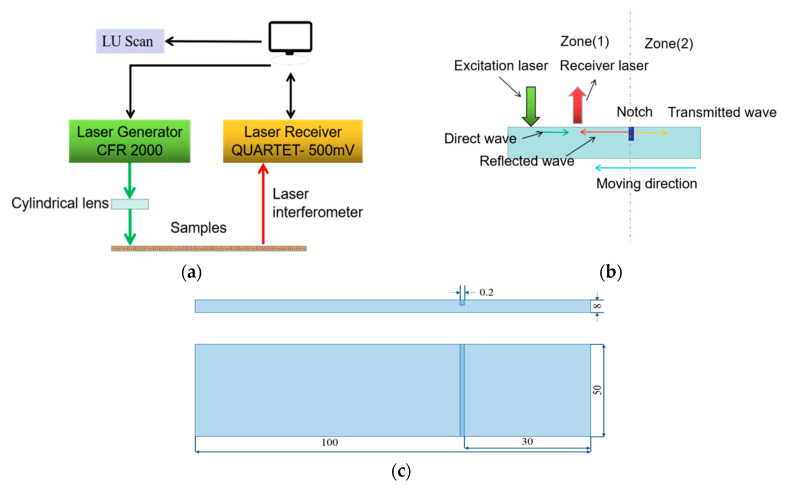
(**a**) Ultrasonic laser detection system. (**b**) Scanning path. (**c**) Sample size (unit: mm).

**Figure 12 sensors-20-05077-f012:**
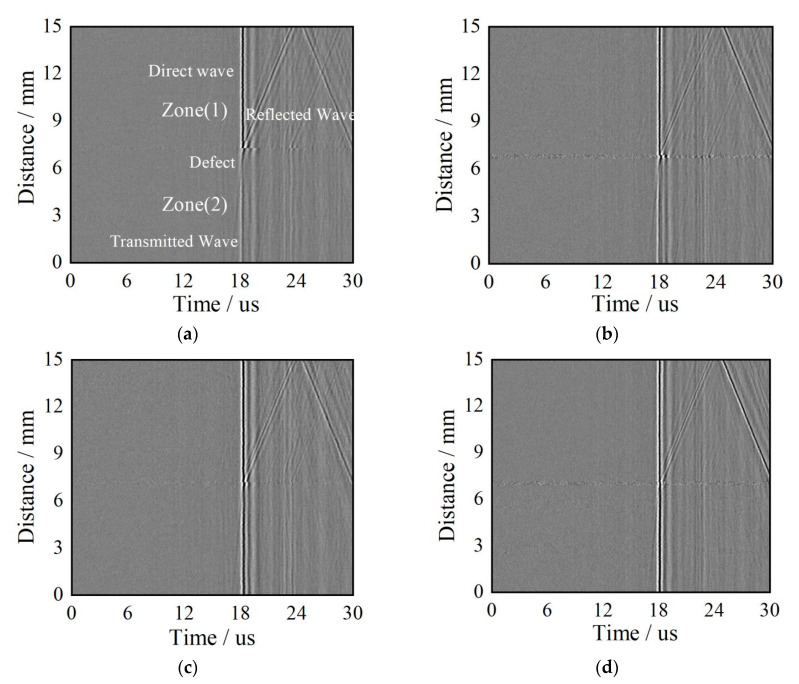
B-scanning images of surface cracks (**a**) The notch depth is 0.5 mm. (**b**) The notch depth is 0.4 mm. (**c**) The notch depth is 0.3 mm. (**d**) The notch depth is 0.2 mm. (**e**) Transmitted signals in the time domain. (**f**) Reflected signals in the time domain.

**Figure 13 sensors-20-05077-f013:**
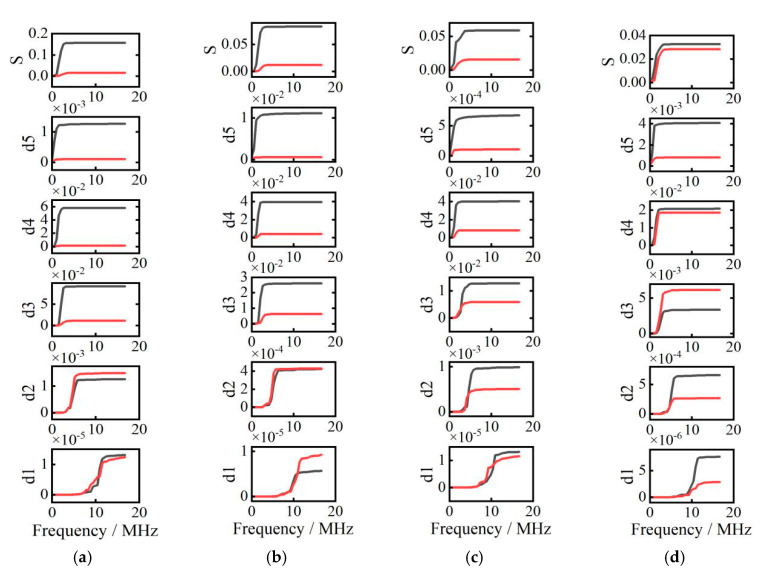
The curves of energy vs. frequency (**a**) crack depth is 0.2 mm, (**b**) crack depth is 0.3 mm, (**c**) crack depth is 0.4 mm, and (**d**) crack depth is 0.5 mm.

**Figure 14 sensors-20-05077-f014:**
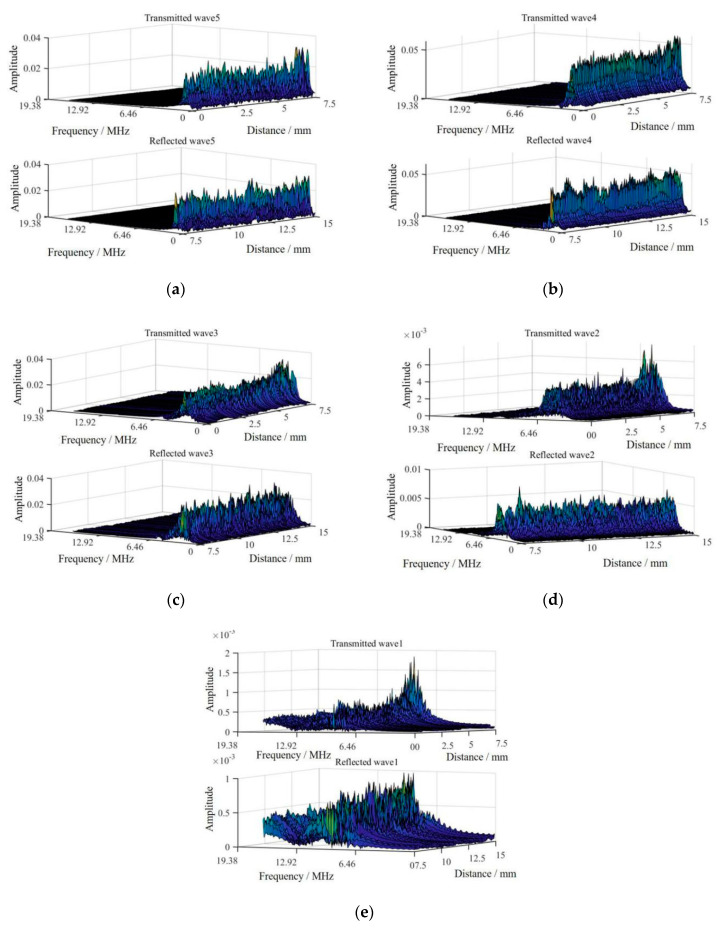
Decomposed signals in frequency-spatial domain. (**a**) Decomposed signals are for d1 signal, (**b**) decomposed signals are for d2 signal, (**c**) decomposed signals are for d3 signal, (**d**) decomposed signals are for d4 signal and (**e**) decomposed signals are for d5 signal.

**Figure 15 sensors-20-05077-f015:**
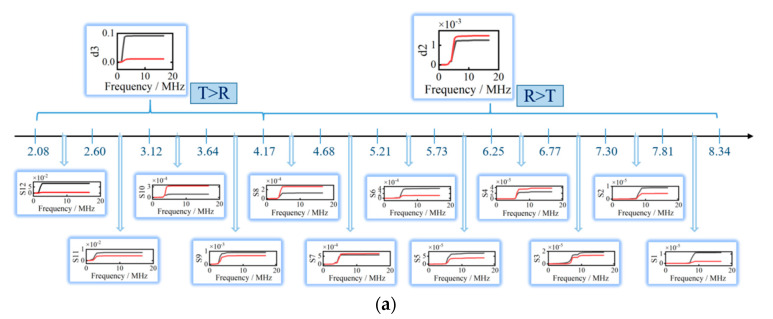
The decomposed frequencies distribution for crack. (**a**) At crack with depth of 0.2 mm. (**b**) At crack with depth of 0.3 mm. (**c**) At crack with depth of 0.4 mm. (**d**) At crack with depth of 0.5 mm.

**Figure 16 sensors-20-05077-f016:**
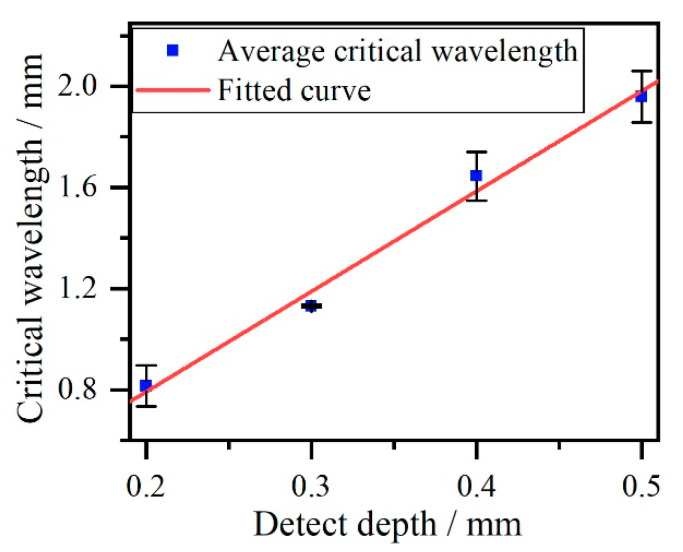
The fitted curve of critical wavelength vs. crack depth.

**Table 1 sensors-20-05077-t001:** The frequency range of decomposed layers.

Layers	d1	d2	d3	d4	d5
Frequency Range /MHz	8.34–16.67	4.17–8.34	2.08–4.17	1.04–2.08	0.52–1.04

**Table 2 sensors-20-05077-t002:** Estimated crack depth and relative error.

Sample	The Critical Frequency/MHz	The Critical Wavelength/mm	Estimated Depth/mm	Error
T = 0.2 mm	3.34	0.883	0.883/4 = 0.221	10.5%
T = 0.3 mm	2.78	1.060	1.060/4 = 0.265	11.7%
T = 0.4 mm	1.89	1.560	1.560/4 = 0.390	2.5%
T = 0.5 mm	1.38	2.136	2.136/4 = 0.534	6.8%

**Table 3 sensors-20-05077-t003:** Estimated crack depth and relative error for simulation element with the maximum size 0.2 mm.

Sample	The Critical Frequency/MHz	The Critical Wavelength mm	Estimated Depth/mm	Error
T = 0.2 mm	3.81	0.776	0.776/4 = 0.194	3%
T = 0.3 mm	2.21	1.334	1.334/4 = 0.334	11.3%
T = 0.4 mm	1.56	1.890	1.890/4 = 0.473	18.25%
T = 0.5 mm	1.45	2.033	2.033/4 = 0.508	1.6%

**Table 4 sensors-20-05077-t004:** Estimated crack depth and relative error for simulation element with the maximum size 0.5 mm.

Sample	The Critical Frequency/MHz	The Critical Wavelength/mm	Estimated Depth/mm	Error
T = 0.2 mm	4.69	0.629	0.629 / 4 = 0.157	21.5%
T = 0.3 mm	2.72	1.084	1.060 / 4 = 0.271	9.6%
T = 0.4 mm	1.56	1.890	1.560 / 4 = 0.473	18.25%
T = 0.5 mm	1.45	2.033	2.136 / 4 = 0.508	1.6%

**Table 5 sensors-20-05077-t005:** Positions for Pair A, B and C.

Name	Position of Reflection Wave Picked	Position of Transmission Wave Picked
A	3.4 mm	4.8 mm
B	5.5 mm	6.5 mm
C	6.5 mm	5.5 mm

**Table 6 sensors-20-05077-t006:** Estimated depth and relative error for A.

Sample	The Critical Frequency/MHz	The Critical Wavelength mm	Estimated Depth/mm	Error
T = 0.2 mm	3.72	0.792	0.792/4 = 0.198	1%
T = 0.3 mm	2.60	1.134	1.134/4 = 0.284	5.33%
T = 0.4 mm	1.75	1.685	1.685/4 = 0.421	5.25%
T = 0.5 mm	1.42	2.076	2.076/4 = 0.519	3.8%

**Table 7 sensors-20-05077-t007:** Estimated depth and relative error for B.

Sample	The Critical Frequency MHz	The Critical Wavelength/mm	Estimated Depth/mm	Error
T = 0.2 mm	3.94	0.748	0.748/4 = 0.187	6.5%
T = 0.3 mm	2.62	1.125	1.125/4 = 0.281	6.33%
T = 0.4 mm	1.92	1.535	1.535/4 = 0.384	5.33%
T = 0.5 mm	1.56	1.900	1.900/4 = 0.475	5%

**Table 8 sensors-20-05077-t008:** Estimated depth and relative error for C.

Sample	The Critical Frequency/MHz	The Critical Wavelength/mm	Estimated Depth/mm	Error
T = 0.2 mm	3.25	0.907	0.907 / 4 = 0.227	13.5%
T = 0.3 mm	2.6	1.134	1.134 / 4 = 0.284	5.33%
T = 0.4 mm	1.72	1.714	1.714 / 4 = 0.429	7.25%
T = 0.5 mm	1.56	1.900	1.900 / 4 = 0.475	5%

**Table 9 sensors-20-05077-t009:** Average values and variance for estimated crack depth.

Sample	Average Critical Wavelength/mm	Average Estimated Crack Depth/mm	Error	Variance
T = 0.2 mm	0.816	0.816/4 = 0.229	2%	0.0821
T = 0.3 mm	1.131	1.131/4 = 0.283	5.67%	0.0052
T = 0.4 mm	1.645	1.645/4 = 0.411	2.75%	0.0961
T = 0.5 mm	1.959	1.959/4 = 0.490	2%	0.1016

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
