# Peer review of "An Approach to Size Sub-Wavelength Surface Crack Measurements Using Rayleigh Waves Based on Laser Ultrasounds"

_sensors, 2020, doi:10.3390/s20185077_

Round 1

Reviewer 1 Report

1、Fig. 13 and Fig. 14 are not shown in the manuscript, I wonder if there is something wrong with the downloaded manuscript.

2、In recent years, some researches have focused on the detection of crack depth based on the critical frequency method. So, what is the main innovation of this paper? I suggest that the author highlight his innovation and the difference between his work and others.

3、I suggest that some recent references  about the last development of laser ultrasonic should be added in the Introduction.

4、Whether the transmitted and reflected waves measured in the experiment contain noise signals? Do these noise signals affect the results? What is the amplitude of the noise signal?

Reviewer 2 Report

This paper shows the application of Rayleigh waves to the detection of cracks (cracks experimentally simulated as artificial notches) in a material. The analysis is both: numerical and experimental. The procedure has novelty and has scientific soundness. I think that this paper would have interest for readers, but it is necessary to enhance some aspects, such as, the english and the references.

The abstract must be reorganized, and the first sentence must be included but in other position (see attached file).

Bibliographic references are not enough. A fast study in "Scholar google" shows a lot of references related to the subject (searching by Rayleigh waves and ulltrasonic laser).

In figure 15, more pints must be included for enhance the statistical significance of the regression.  

There are serious mistakes associated to grammar, the english must be revised by a professional or an english-native speaker. 

In the attached file, some grammar mistakes are indicated. 

Reviewer 3 Report

The authors analyze wave scattering reflection and transmission generated by a laser source upon incidence on a crack. The work contains both theoretical and experimental parts. It allows for the assessment of the relation characterizing the crack depth and critical wavelength, which is experimentally validated to a satisfactory extend. The work is overall well-structured and provides insights and results of practical significance. However, there exist some important issues that need to be addressed. In particular:

  1. Important simulation details are missing from FEM analysis of Section 3. What is the type of elements used? What is the sensitivity of the solution to the mesh size used? Is the solution converging for the mesh size used and the type appropriate?
  2. Relevant to the previous comment comes the result of Fig. 7. The transmitted wave follows a descending branch for the Gaussian function approximation, while the FEM results appear to be constant. Moreover, are the points provided for the 2 cases applied to the same crack depth values? The center frequency seems to be considerably displaced.
  3. A more concrete explanation of the theoretical rational employed needs to be provided. In particular: How is the number of necessary wavelet transformations employed selected? How is the initial search space determined? Relevant comments are provided after Fig. 13, but no methodology is explicated.
  4. Better summary/presentation of the results is required at different positions. Specifically, the legends and subplots of Fig. 7 are difficult to follow. The same applies to Fig. 8, whose labels should be beforehand summarized in a Table for conciseness and clarity and not first commented after Fig. 8.
  5. The results of Fig 9 and 15 indicate that a linear relation between the critical wavelength and the defect depth. Can a linear relation be physically justified/explained? The fitting points are rather few (4 and 3 accordingly). A more refined grid should be provided, desirably upon a 0.05/mm depth. What are the bounds of critical wavelength/ crack depth that the method can trustworthily capture? Relevant information needs to be given.
  6. More detailed information with respect to the computational cost of the method needs to be provided. What is the frequency range/grid size that needs to be tested for critical frequency to be determined?
  7. The theoretical methodology explained in section 2 is based on a frequency independent amplitude response. However, several recent studies have shown that a wave-amplitude and frequency dependent response can be physically relevant for certain frequency ranges, leading to the need for a non-linear waveform formulation (see org/10.1016/j.jsv.2019.05.011 or doi.org/10.1016/j.jsv.2018.06.006). The linearity of the waveform solution should be highlighted in the introduction section of the manuscript. The literature review should be extended and clearly distinguished from possible nonlinear effects of the kind, here not considered.
  8. In page 16 line 325, the results discussed refer to Figure 12 or 14?
  9. The language of the manuscript presents substantial shortcomings at different positions. Articles are missing and erroneous sentences. Indicatively line 156, page 5 should be: In this work, the depth of the detected crack… than the wavelength… The manuscript should be thoroughly proof read by a language expert and the corresponding shortcomings need to be corrected.

Round 2

Reviewer 1 Report

In my opinion, this paper has been weel revised, so my recommendation is accept. 

Reviewer 2 Report

Suggested creation were made. English was improved. Figure 15 was explained.

I recommend to publish the article.

Reviewer 3 Report

The authors have extensively revised the manuscript, addressing all comments raised and substantially improving its quality.Theoretical and presentation shortcomings have been addressed, while the important conclusions of the study are now clearly elaborated and highlighted. I suggest that the paper is published in its current form.